

# Baseline gene expression in subcutaneous adipose tissue predicts diet-induced weight loss in individuals with obesity

Ali Oghabian[1],*, Birgitta W. van der Kolk[1],*, Pekka Marttinen[2], Armand Valsesia[3], Dominique Langin[4,5], W. H. Saris[6], Arne Astrup[7], Ellen E. Blaak[6] and Kirsi H. Pietiläinen[1,8]

[1] Obesity Research Unit, Research Program for Clinical and Molecular Metabolism, Faculty of Medicine, University of Helsinki, Helsinki, Finland
[2] Helsinki Institute for Information Technology HIIT, Department of Computer Science, Aalto University, Espoo, Finland
[3] Nestlé Institute of Health Sciences, Lausanne, Switzerland
[4] Department of Biochemistry, Toulouse University Hospitals, Toulouse, France
[5] Institut des Maladies Métaboliques et Cardiovasculaires, I2MC, Université de Toulouse, Inserm, Université Toulouse III—Paul Sabatier (UPS), Toulouse, France
[6] Department of Human Biology, NUTRIM School of Nutrition and Translational Research in Metabolism, Maastricht University, Maastricht, The Netherlands
[7] Healthy Weight Center, Novo Nordisk Fonden, Copenhagen, Denmark
[8] Healthy Weight Hub, Abdominal Center, Helsinki University Hospital and University of Helsinki, Helsinki, Finland
* These authors contributed equally to this work.

Corresponding author
Ali Oghabian,
ali.oghabian@helsinki.fi

## ABSTRACT

**Background:** Weight loss effectively reduces cardiometabolic health risks among people with overweight and obesity, but inter-individual variability in weight loss maintenance is large. Here we studied whether baseline gene expression in subcutaneous adipose tissue predicts diet-induced weight loss success.

**Methods:** Within the 8-month multicenter dietary intervention study DiOGenes, we classified a low weight-losers (low-WL) group and a high-WL group based on median weight loss percentage (9.9%) from 281 individuals. Using RNA sequencing, we identified the significantly differentially expressed genes between high-WL and low-WL at baseline and their enriched pathways. We used this information together with support vector machines with linear kernel to build classifier models that predict the weight loss classes.

**Results:** Prediction models based on a selection of genes that are associated with the discovered pathways 'lipid metabolism' (max AUC = 0.74, 95% CI [0.62–0.86]) and 'response to virus' (max AUC = 0.72, 95% CI [0.61–0.83]) predicted the weight-loss classes high-WL/low-WL significantly better than models based on randomly selected genes ($P < 0.01$). The performance of the models based on 'response to virus' genes is highly dependent on those genes that are also associated with lipid metabolism. Incorporation of baseline clinical factors into these models did not noticeably enhance the model performance in most of the runs. This study demonstrates that baseline adipose tissue gene expression data, together with supervised machine learning, facilitates the characterization of the determinants of successful weight loss.

# INTRODUCTION

Weight loss resulting from a low-calorie diet (LCD) effectively improves obesity and reduces the health risks among people with overweight and obesity (*Bray et al., 2016*). A moderate body weight reduction of 5–10% induced through dieting significantly reduces disease risk and improves the metabolic profile (*Gaal, Wauters & Leeuw, 1997*). However, inter-individual variability in weight loss in response to lifestyle interventions is large and long-term weight maintenance remains rather poor (*Anderson et al., 2001*). Identifying markers that can predict the responsiveness to various weight loss interventions is significant for clinical care.

In recent years, several studies have applied machine learning methods in the obesity field (for extensive review see *Safaei et al., 2021*) and a few studies have attempted to predict weight variations in humans by applying machine learning methods. These machine learning approaches help us better understand the biological complexity underlying inter-individual variability in weight loss. The prediction studies have mostly focused on energy intake and expenditure (*Thomas et al., 2011*, *2014*), macronutrient balance (*Chow & Hall, 2008*), anthropometrics (*Finkler, Heymsfield & St-Onge, 2012*), and glycemic and insulinemic statuses (*Ritz et al., 2019*; *Valsesia et al., 2020*), blood DNA methylation (*Keller et al., 2020*) as predictor variables. Another attractive factor to predict weight loss is based on adipose tissue function (*Goossens, 2017*) as adipose tissue is a main regulator for post-dieting physiology (*MacLean et al., 2015*), responds dynamically to caloric intake (*van Baak & Mariman, 2019*), and weight loss leads to a desired reduction in adiposity.

Indeed, a few studies have utilized supervised machine learning on subcutaneous adipose tissue gene expression data to predict weight-loss response to dietary interventions (*Mutch et al., 2011*; *Armenise et al., 2017*; *Valsesia et al., 2020*). But these studies mostly focused on glycemic control responders following weight loss (*Valsesia et al., 2020*) or they used the adipose tissue gene expression response to caloric restriction for the prediction models (*Mutch et al., 2011*; *Armenise et al., 2017*). Overall, these studies emphasize the importance of adipose tissue metabolism in weight loss interventions and its potential in weight loss predictions.

The current work aims to identify whether baseline gene expression in abdominal subcutaneous adipose tissue can predict the weight loss status following a 2-month low-calorie diet (LCD) and a subsequent 6-months weight maintenance period. We reanalyzed retrospectively baseline RNA-seq data from individuals from the previously published Diet Obesity and Genes (DiOGenes) study (*Larsen et al., 2010*). This study provides important knowledge on baseline adipose tissue metabolism that may prove beneficial in the long term for the development of personalized strategies to improve successful weight maintenance.

## MATERIALS AND METHODS

### Study design

DiOGenes is a multicenter, randomized, dietary intervention study (trial number NCT00390637), described in detail elsewhere (*Larsen et al., 2010*). Briefly, 938 healthy adults with overweight or obesity (BMI 27–45 kg/m$^2$) without concomitant medications entered the first phase of the study: a 2-month weight-loss period using a complete meal replacement low calorie diet (LCD) providing 800 kcal/day (Modifast; Nutrition et Santé France, Revel, France). Among the 781 participants who completed the LCD period, 773 achieved >8% weight loss and those individuals were randomly assigned to one of five different weight maintenance diets for the next 6 months, differing in macronutrient compositions (a 2 × 2 factorial combination diet of low/high protein and a low/high glycemic index or a control diet). A total of 548 (71%) participants completed the whole 8-month intervention. Here, we examined data from 281 out of these 548 participants, for whom abdominal subcutaneous adipose tissue RNA sequencing data at baseline was available. Weight development and metabolic health were assessed at baseline, 2, and 8 months.

The ethics committee of each center/country approved the protocol. The committees included (1) Medical ethical commission from Maastricht University, NL (2) Copenhagen ethical research commission, DK (3) Bedfordshire local Research Ethics Committee, Luton and Dunstable Hospital NHS Trust, UK (4) Ethics Committee of the Faculty Hospital, Prague University, CZ (5) Ethical Commission by NMTI, Sofia, BG (6) Ethical Commission University Potsdam, D (7) Ethical Commission Medical University, Navarra, SP (8) Scientific council Heraklion general university hospital, Heraklion, GR and (9) Commission Cantonale d' éthique de la recherche sur l' être humain, Canton de Vaud, CH. Furthermore, the protocol was in accordance with the Declaration of Helsinki. All study participants provided written consent.

### Subcutaneous adipose tissue biopsy and transcriptomic analyses

Abdominal subcutaneous adipose tissue needle biopsies were collected 6–8-cm laterally from the umbilicus under local anesthesia. Thereafter, biopsy specimens were snap-frozen in liquid nitrogen and stored at −80 °C until further analysis. Total RNA was extracted and analyzed from subcutaneous adipose tissue biopsy specimens as described elsewhere (*Armenise et al., 2017*; *van der Kolk et al., 2019*). For each sample, gene expression was then examined using 100-nucleotide-long paired-end RNA sequencing with an Illumina HiSeq 2000 of libraries prepared using the Illumina TruSeq kit according to the manufacturer's instructions. We used GenomicAlignments to retrieve the number of reads mapping onto 53,343 genes (GRCh37 assembly). Only reads with both ends mapping onto coordinates of a single gene were considered (*Armenise et al., 2017*; *van der Kolk et al., 2019*). The data is accessible through Gene Expression Omnibus under the accession no. GSE95640.

### Construction, training and testing of weight loss prediction models

A diagram illustrating our method is shown in Figs. 1A and 1B. The RNAseq data from the studied individuals at baseline were split into training and testing subsamples. To prevent
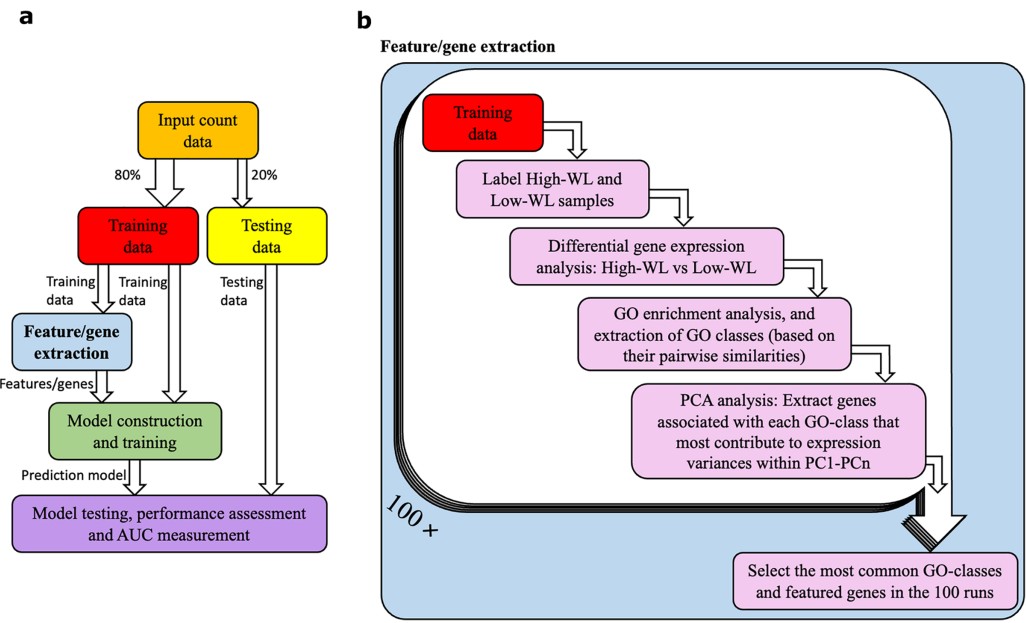

**Figure 1  Data analysis workflow.** (A) Abstract illustration of the analysis performed during a single run (out of 100). Note that for each analysis (*i.e.*, feature/gene extraction, and construction, training and testing of the prediction models) the necessary normalization and adjustment for sex, age and study center effects are carried out. (B) The "feature/gene extraction" precedes all other analysis (*i.e.*, prediction model construction, training and testing) and is shown in further details. In each run the corresponding training data is analyzed. The outputs during the 100 runs are considered to extract the significant GO categories and genes that are used for construction of prediction models. Furthermore, the dimensions PC1-PCn collectively explain 70% of the gene expression variance.

overfitting of the prediction models, we used a five-fold cross-validation with 100 resampling (*i.e.*, 100 times our samples were randomly divided to 80% training and 20% testing subsets). This machine learning setup was used for all the prediction models that are reported in this article. Within each training and testing subset, participants were divided into high-weight losers (WL) and low-WL groups based on the cutoff of median weight loss percentage in the subset.

We used support vector machines (SVM) with linear kernel (and cost value of 1) to build classifier prediction models. Prior to the training/testing of the models, we applied the *ComBatSeq()* function supported by the SVA R library (*Johnson, Li & Rabinovic, 2006*; *Zhang, Parmigiani & Johnson, 2020*) on the count matrix to adjust for biases and effects introduced by sex, age and study center. We then applied $Log_2$ Count Per Million (LCPM) scaling while adjusting the library sizes by normalization factors that were calculated using the Trimmed Mean of M values (TMM) (*Robinson, McCarthy & Smyth, 2010*). The normalizations were carried out independently for the 80% training and the remaining 20% testing sub-samples. Furthermore, the training and testing of the prediction models were carried out on their corresponding sub-samples. We finally used the ROCR R-package to measure the area under the receiver operating characteristic (ROC) curve, *i.e.*, AUC. The 95% confidence interval (CI) of the AUC of the most optimal models were

calculated using the DeLong method supported by the pROC R package (*DeLong, DeLong & Clarke-Pearson, 1988*).

To ensure that the performances of the models are not affected by unaccounted for biases within the expression data, the performances of all prediction models (based on the studied gene-lists, see below) were compared to those that featured randomly selected genes as their input parameters. In further detail, we created 100 lists of randomly picked genes (each list featuring ten genes) over the whole genome, from the genes with >5 mapped reads across all samples. Next, we built 100 prediction models based on the randomly selected genes. For each prediction model a *P*-value was measured. The *P*-value is the fraction of the 100 prediction models that were based on the randomly picked genes that resulted in higher median AUCs compared to the prediction model (*i.e.,* based on a studied gene-list). We considered $P < 0.05$ as significant.

## Selection of genes for the prediction models

We studied differentially expressed genes (DEGs), comparing high-WL to low-WL, by applying DESeq2 on the training samples in each cross-validation run (*Love, Huber & Anders, 2014*). We adjusted the models for sex, age and study center and we considered the nominal $P < 0.01$ as significant. We then extracted the biological process Gene Ontology (GO) categories enriched in the significant DEGs, using the *enrichGO()* function supported by the *clusterProfiler* R/Bioconductor package (with the default parameter settings of $P < 0.05$ and $q < 0.2$ cutoffs) (*Yu et al., 2012*). This function applies hypergeometric distribution to calculate *P*-values based on the number of selected genes associated with the GO category, the overall number of genes associated with the GO category, and the overall number of genes in background (*Boyle et al., 2004*).

The Benjamini Hochberg method was used to correct for multiple testing (*Benjamini & Hochberg, 1995*) and we considered FDR <0.05 significant. Furthermore, we constructed enrichment maps based on the similarities of the ontologies using the *enrichplot* R package and the redundant GO categories were filtered based on a ≥70% similarity cutoff.

We applied the Jaccard similarity coefficient method supported by the *pairwise_termsim()* function of the *enrichplot* R package to construct the similarity matrix of the GO categories (*Yu & Hu, 2020*). The 'components' within the enrichment maps, *i.e.,* biological process GO categories that for each pair at least one path exists, were finally extracted. We used the vertices within these components, *i.e.,* classes of interrelated biological process GO categories, to extract the suitable genes for construction of the prediction models.

The biological process GO categories discovered from comparing the high-WL *vs.* low-WL include fatty acid metabolic pathways (referred to as Lipid), mitotic/meiotic pathways (referred to as Mitosis), and 'response to virus' pathways (referred to as Virus). Expression data of the genes that are associated with a pathway (*i.e.,* Lipid, Mitosis or Virus) were extracted and principle component analysis (PCA) was performed. From each list of genes associated with a pathway, ten genes that most contributed to the PC dimensions that collectively explained ≥70% of the variance within the expression data was chosen. Since we used five-fold cross-validation with 100 resampling, the training data varied across the different runs and the discovered genes from a specific pathway across the

different runs somewhat varied too. Therefore, from each studied Lipid/Mitosis/Virus gene list, the ten most frequently featured genes during the 100 runs were selected as input parameters to the corresponding prediction models. Finally, to further analyze the performance of the prediction models based on lipid metabolism and 'response to virus' genes, we also constructed models based on combined gene lists (named as Lipid + Virus), and prediction models based on genes featured in one but absent from the other (*i.e.*, Lipid —Virus and Virus—Lipid).

### Incorporating clinical factors into the prediction models

We studied the effects of anthropometric and clinical factors (Table S1 and S2) by building a prediction model solely based on these factors as well as incorporating them into the input parameters of the Lipid and Virus models. The values of the clinical factors were adjusted for sex and age using the non-parametric method supported by *ComBat()* function of the *sva* R/Bioconductor package (*Johnson, Li & Rabinovic, 2006*; *Zhang, Parmigiani & Johnson, 2020*). The highly skewed clinical data (*i.e.*, skewness > 1 or skewness < −1) were $\log_e$ scaled. For each Lipid and Virus model that incorporated a clinical factor, a *P*-value was calculated by using paired t-test, by comparing their performances to the original Lipid/Virus prediction models that lack the clinical factor information. We considered $P < 0.05$ as significant.

### Expression heatmaps

We normalized the gene expression within all samples across all three time points using variance stabilizing transformation (VST) supported by DESeq2, whilst adjusting for sex, age and study center in the design model (*Anders & Huber, 2010*).

## RESULTS

### Baseline characteristics and overall clinical outcome

Baseline characteristics of the participants are described in Table 1. For descriptive reasons, we created this table based the median weight loss percentage achieved at 8-months (9.9%) from 281 participants. Sixty-five percent of the participants were females, and the participants were on average 42-years-old. At baseline, the BMI of included participants was $34.6 \pm 0.2$ kg/m$^2$. The high-WL and low-WL groups were comparable at baseline in term of weight and insulin sensitivity (Table 1). Upon LCD at the 2-month time point, both groups had achieved significant weight loss (>8% initial body weight); yet the high-WL group had significantly higher weight loss (mean difference 2.3%, $P < 0.001$). By design, following 8 months, the high-WL group lost significantly more of their initial body weight $15.4 \pm 4.6\%$ compared to the low-WL group who lost $5.8 \pm 3.4\%$ ($P < 0.001$).

### Prediction models based on biological processes associated with weight-loss

To create a predictive model for diet-induced weight loss success based on global subcutaneous adipose tissue gene expression and biological processes, we extracted the significantly DEGs in the training subsamples used in each cross-validation run by

**Table 1 Participant characteristics.**

| | High-WL baseline | 2 Months | 8 Months | Low-WL baseline | 2 Months | 8 Months | *P*-value baseline |
|---|---|---|---|---|---|---|---|
| Female sex (%) | 64.5 | | | 65.7 | | | 0.836 |
| Age (y) | 42.2 ± 6.7 | | | 42.5 ± 5.7 | | | 0.685 |
| Weight loss (%) | | 12.3 ± 2.9 | 15.4 ± 4.6 | | 10.0 ± 1.9* | 5.8 ± 3.4[#] | |
| Weight (kg) | 100.4 ± 18.3 | 87.9 ± 15.9 | 84.8 ± 15.8 | 97.7 ± 16.7 | 88.0 ± 15.1 | 92.0 ± 15.7 | 0.211 |
| BMI (kg/m$^2$) | 34.6 ± 4.8 | 30.3 ± 4.3 | 29.2 ± 4.2 | 34.0 ± 4.4 | 30.6 ± 4.1 | 32.0 ± 4.1 | 0.291 |
| Cholesterol (mmol/L) | 4.8 ± 0.9 | 4.2 ± 0.8 | 4.8 ± 0.8 | 4.9 ± 1.1 | 4.3 ± 1.0 | 5.0 ± 1.0 | 0.339 |
| LDL (mmol/L) | 3.0 ± 0.8 | 2.5 ± 0.7 | 2.9 ± 0.8 | 3.1 ± 0.9 | 2.6 ± 0.8 | 3.0 ± 0.9 | 0.479 |
| HDL (mmol/L) | 1.2 ± 0.3 | 1.2 ± 0.3 | 1.4 ± 0.3 | 1.3 ± 0.3 | 1.2 ± 0.3 | 1.4 ± 0.3 | 0.582 |
| TAG (mmol/L) | 1.4 ± 0.6 | 1.1 ± 0.5 | 1.1 ± 0.5 | 1.4 ± 0.7 | 1.1 ± 0.4 | 1.4 ± 0.6 | 0.682 |
| NEFA (µmol/L) | 680 ± 294 | 747 ± 231 | 574 ± 229 | 640 ± 281 | 685 ± 218 | 564 ± 227 | 0.295 |
| Glucose (mmol/L) | 5.1 ± 0.6 | 4.8 ± 0.5 | 4.9 ± 0.5 | 5.1 ± 0.6 | 4.9 ± 0.5 | 5.1 ± 0.5 | 0.734 |
| Insulin (mU/L) | 12.6 ± 18.1 | 9.2 ± 15.1 | 9.9 ± 14.3 | 10.9 ± 6.5 | 7.8 ± 3.9 | 9.1 ± 4.5 | 0.312 |
| HOMA-IR | 3.4 ± 5.0 | 2.2 ± 3.9 | 2.5 ± 4.0 | 2.9 ± 1.9 | 1.9 ± 1.0 | 2.3 ± 1.3 | 0.358 |

**Note:**
Data are reported as mean ± SD. *P* values for baseline comparisons were obtained by using the Pearson $c^2$ test for sex, and by unpaired *t*-test for the other variables (*$P < 0.001$ different from 2 months, [#]$P < 0.001$ different from 8 months). Skewed variables were log$_e$-transformed before analysis. BMI, body mass index; HDL, high-density lipoprotein; LDL, low-density lipoprotein; TAG, triacylglycerol; NEFA, non-esterified fatty acids; HOMA-IR, homeostatic model for the assessment of insulin resistance. Data based on $n = 281$.

comparing the high-WL to the low-WL subsamples at baseline. Subsequently, we created GO enrichment maps that illustrate the relationships of the discovered significantly enriched biological process GOs based on their pairwise gene overlaps. Throughout the 100 runs, the most frequently discovered biological process GO categories were associated with fatty acid metabolic pathways (*i.e.*, Lipid, discovered in 42 runs), mitosis/meiosis pathways (Mitosis, discovered in 36 runs), and 'response to virus' pathways (Virus, discovered in 27 runs) (Table 2). In 34 runs, however, no significant GOs were discovered. The largest nonoverlapping Lipid, Mitosis, and Virus associated biological process classes discovered across the 100 cross validation runs are illustrated in Fig. 2A.

Next, we extracted ten genes from each of the three discovered pathways (Fig. 2B and Table 3). We used the baseline expression of these genes as input parameters to construct weight-loss prediction models. The Lipid model (median AUC = 0.58, *P* < 0.01) (Fig. 3A) and Virus model (median AUC = 0.59, *P* < 0.01) (Fig. 3C) predicted the 8-months weight loss of the individuals in our study significantly better than 100 models that feature randomly selected genes as their input parameters or the Mitosis model (median AUC = 0.50, *P* = 0.21) (Fig. 3B). The maximum AUC achieved by the Lipid models was 0.74 with 95% CI [0.62–0.86], whilst for the Virus models was 0.72 with 95% CI [0.61–0.83] and for the Mitosis models was 0.66 with 95% CI [0.54–0.78].

Notably, the genes in the Virus model were highly in common with the genes in the Lipid models (Fig. 2B) indicating that the discovered 'response to virus' pathways have genes with heterogenous functions that are also associated with the lipid metabolism pathways. Therefore, we further examined prediction models that feature combinations of the Lipid and Virus genes. Prediction models based on combined 'response to virus' and

**Table 2 GO classes and their descriptions.** Descriptions of the pathway classes with which the genes used in the weight loss prediction models are associated.

| Class name | Description |
|---|---|
| Lipid | Fatty acid metabolic process, fatty acid derivative metabolic process, long-chain fatty acid metabolic process, fatty acid derivative biosynthetic process, fatty-acyl-CoA biosynthetic process, long-chain fatty acid biosynthetic process, acyl-CoA metabolic process, thioester metabolic process, nucleoside bisphosphate metabolic process, purine nucleoside bisphosphate metabolic process, ribonucleoside bisphosphate metabolic process, acyl-CoA biosynthetic process, demethylation, thioester biosynthetic process, triglyceride biosynthetic process, response to xenobiotic stimulus, retinoic acid metabolic process, retinol metabolic process, steroid biosynthetic process, secondary alcohol biosynthetic process, xenobiotic metabolic process, oxidative demethylation, steroid metabolic process, drug metabolic process, regulation of lipid metabolic process, and ribose phosphate metabolic process. |
| Mitosis | Meiotic sister chromatid cohesion, meiotic sister chromatid segregation, meiotic cell cycle, chromosome segregation, nuclear chromosome segregation, nuclear division, regulation of chromosome segregation, regulation of mitotic nuclear division, regulation of nuclear division, sister chromatid segregation, mitotic nuclear division, chromosome localization, establishment of chromosome localization, mitotic cell cycle checkpoint, regulation of chromosome organization, regulation of mitotic cell cycle phase transition, chromosome condensation, microtubule-based movement, regulation of chromosome separation, cytokinesis, cell cycle G2/M phase transition, G2/M transition of mitotic cell cycle, cholesterol metabolic process, mitotic sister chromatid segregation, mitotic spindle assembly checkpoint, regulation of sister chromatid segregation, drug catabolic process, hormone metabolic process, mitotic cytokinesis, and secondary alcohol metabolic process. |
| Virus | Response to virus, negative regulation of viral genome replication, defense response to virus, and estrogen metabolic process |

lipid metabolism gene-lists (Lipid + Virus) did not perform significantly better than Virus or Lipid models (Figs. S1A, S1B). Furthermore, excluding the genes that are annotated with 'response to virus' from the Lipid models (Lipid–Virus) did not affect the performance of the Lipid metabolism significantly (Fig. S1C and Table S3). However, prediction models that featured genes associated with 'response to virus' but not annotated with lipid metabolism (Virus–Lipid) performed drastically different compared to the Virus models (Fig. S1D). The performance of these models was no longer significantly better than those models that feature randomly selected genes (median AUC = 0.51, max AUC = 0.67, 95% CI [0.51–0.75], $P = 0.16$) (Fig. S2A and Table S3). This shows that the performance of the Virus models is highly dependent on the genes that they have in common with the Lipid models. In contrast, the performance of the Lipid models are not dependent on the genes that they have in common with the Virus models (Fig. S2B and Table S3).

To unravel the conditions under which the Lipid and Virus prediction models best performed, and to explain their bimodal AUC distributions (Figs. 3A, 3C), we compared the training subsamples in the runs that resulted the AUCs higher than 0.60 *vs.* those in the runs that resulted lower AUCs. Note that the overall weight loss during the 8-months correlated highly with the weight loss during the 6-months weight maintenance phase ($r = 0.90$, $P < 2.2e−16$, Pearson correlation test) (Fig. 3D), but our results show that the models perform worst when the training dataset include more samples from individuals who gained weight during the 6-months weight maintenance phase despite being in the high-WL group. In detail, 30 high-WL individuals gained weight during the 6-months weight maintenance of which seven samples regained more than 4% (*i.e.,* 5[th] percentile) weight from the LCD-induced weight loss (Fig. 3D). We realized that runs with AUCs higher than 0.6 included significantly less samples from those seven individuals ($P < 0.05$,

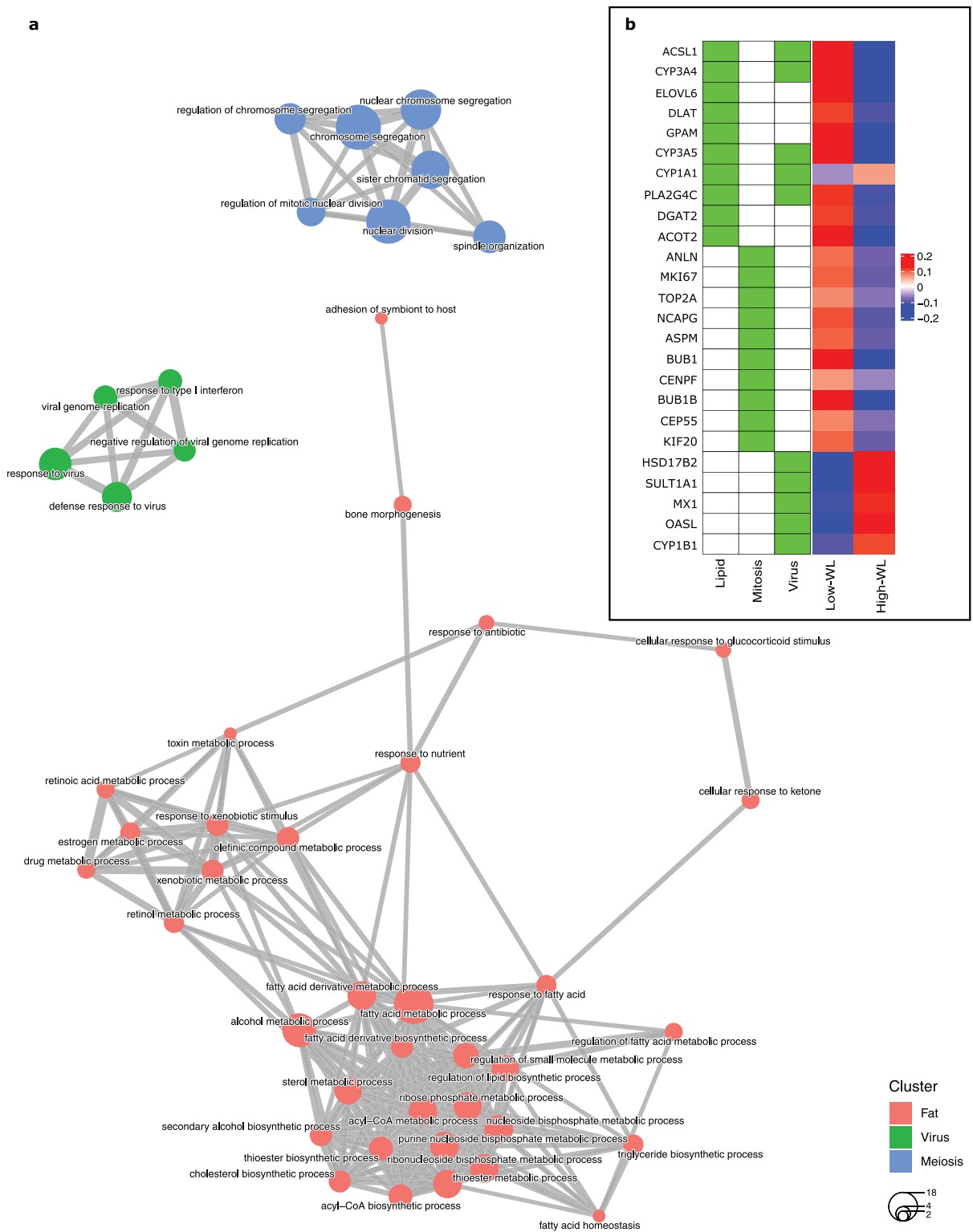

**Figure 2 Baseline biological processes and the gene expression associated with weight loss status.** (A) Biological processes (GO categories) enriched in genes significantly differentially expressed in high-WL individuals *vs.* low-WL individuals. The biological process categories with ≥0.2 measured similarity are connected *via* an edge. The graph illustrates the largest non-overlapping class of biological processes associated with Lipid

**Figure 2** (continued)
Metabolism, Mitosis, and Virus that was achieved throughout the 100 cross-validation runs. (B) Heatmap of the variance stabilizing transformation (VST) normalized read counts (at baseline), adjusted by age, sex and study center of the genes that were used in the Lipid Metabolism, Mitosis, and Virus weight-loss prediction models (scaled by row). The heatmap is plotted for the Low-WL and the High-WL individuals. The table on the left shows the annotations of the genes and the prediction models in which the expression of these genes where used as input parameters.

**Table 3 The genes used in the prediction models.** The ENSEMBL IDs, names, Entrez description of the genes that are used in the prediction models. The names of the prediction models in which are used, or in other words the GO classes with which they are associated, are listed under "GO classes" column. The percentage of the runs in which these genes are discovered to be significantly differentially expressed (when comparing high-WL *vs.* low-WL) are listed in the column "Significant %".

| Gene name | ENSEMBL ID | Entrez gene description | GO classes | Significant % |
|---|---|---|---|---|
| ACOT2 | ENSG00000119673 | Acyl-CoA thioesterase 2 | Lipid | 94 |
| CYP3A5 | ENSG00000106258 | Cytochrome P450, family 3, subfamily A, polypeptide 5 | Lipid, virus | 92 |
| ELOVL6 | ENSG00000170522 | ELOVL fatty acid elongase 6 | Lipid | 92 |
| PLA2G4C | ENSG00000105499 | Phospholipase A2, group IVC (cytosolic, calcium-independent) | Lipid, virus | 82 |
| CYP3A4 | ENSG00000160868 | Cytochrome P450, family 3, subfamily A, polypeptide 4 | Lipid, virus | 79 |
| SULT1A1 | ENSG00000196502 | Sulfotransferase family, cytosolic, 1A, phenol-preferring, member 1 | Virus | 74 |
| DLAT | ENSG00000150768 | Dihydrolipoamide S-acetyltransferase | Lipid | 70 |
| ACSL1 | ENSG00000151726 | Acyl-CoA synthetase long-chain family member 1 | Lipid, virus | 67 |
| HSD17B2 | ENSG00000086696 | Hydroxysteroid (17-beta) dehydrogenase 2 | Virus | 64 |
| NCAPG | ENSG00000109805 | Non-SMC condensin I complex, subunit G | Mitosis | 63 |
| GPAM | ENSG00000119927 | Glycerol-3-phosphate acyltransferase, mitochondrial | Lipid | 63 |
| OASL | ENSG00000135114 | 2′–5′-oligoadenylate synthetase-like | Virus | 61 |
| BUB1B | ENSG00000156970 | BUB1 mitotic checkpoint serine/threonine kinase B | Mitosis | 60 |
| MX1 | ENSG00000157601 | Myxovirus (influenza virus) resistance 1, interferon-inducible protein p78 (mouse) | Virus | 56 |
| ANLN | ENSG00000011426 | Anillin, actin binding protein | Mitosis | 53 |
| CYP1A1 | ENSG00000140465 | Cytochrome P450, family 1, subfamily A, polypeptide 1 | Lipid, virus | 50 |
| MKI67 | ENSG00000148773 | Marker of proliferation Ki-67 | Mitosis | 50 |
| KIF20A | ENSG00000112984 | Kinesin family member 20A | Mitosis | 49 |
| DGAT2 | ENSG00000062282 | Diacylglycerol O-acyltransferase 2 | Lipid | 48 |
| ASPM | ENSG00000066279 | Asp (abnormal spindle) homolog, microcephaly associated (Drosophila) | Mitosis | 47 |
| TOP2A | ENSG00000131747 | Topoisomerase (DNA) II alpha 170 kDa | Mitosis | 47 |
| BUB1 | ENSG00000169679 | BUB1 mitotic checkpoint serine/threonine kinase | Mitosis | 45 |
| CEP55 | ENSG00000138180 | Centrosomal protein 55 kDa | Mitosis | 43 |
| CYP1B1 | ENSG00000138061 | Cytochrome P450, family 1, subfamily B, polypeptide 1 | Virus | 40 |
| CENPF | ENSG00000117724 | Centromere protein F, 350/400 kDa | Mitosis | 38 |

Jonckheere–Terpstra test) in their training sets compared to other runs. This was true for both, Lipid (*P* = 0.048, Jonckheere–Terpstra test) and Virus (*P* = 0.034, Jonckheere–Terpstra test) prediction models. These results indicate that training the models on data from high-WL individuals who regained weight considerably can lead to less efficient prediction models.

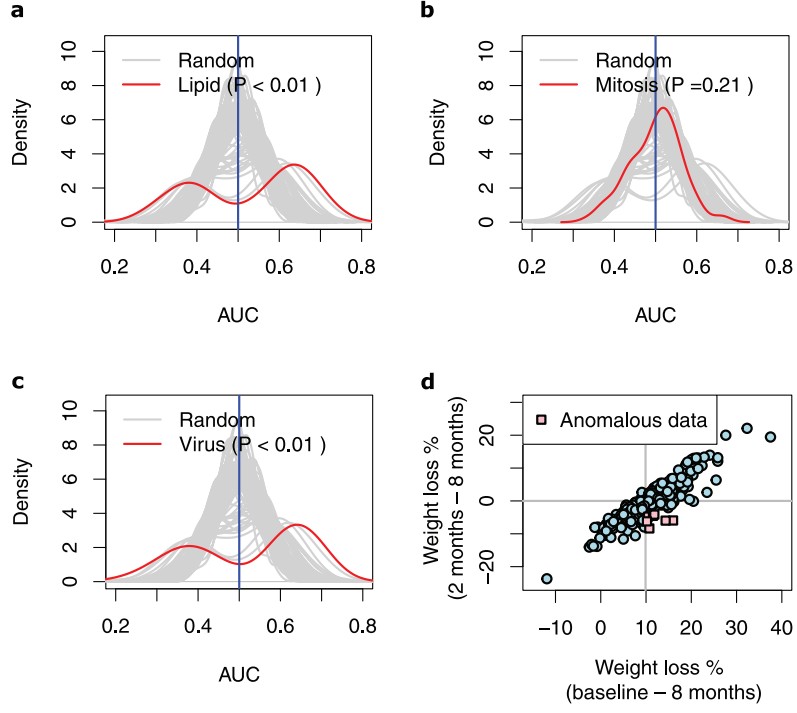

**Figure 3 Performance of the prediction models based on biological processes associated with weight loss.** (A–C) Density plots illustrating the area under ROC curve (AUC) measurements of lipid metabolism, mitosis and virus prediction models across the 100 cross-validation runs. The gray lines correspond to the AUC measurements for 100 prediction models constructed of randomly selected genes. The *P*-value describes the probability that the models based on the randomly picked genes performs better than the models that feature the Lipid/Mitosis/Virus genes as their input parameters. The vertical line at 0.5 AUC indicates the expected performance for random prediction. (D) Scatter plot illustrating weight loss during the whole dietary intervention (baseline—8 months) and during the weight maintenance phase (2–8 months). The individuals who had substantially regained weight (weight loss < −4%) during the weight maintenance phase, despite overall losing weight more than median (anomalous data) are shown with pink squares. The horizontal line shows zero weight loss during the weight maintenance phase whilst the vertical line shows the median of overall weight loss during the 8-months intervention.

## Incorporating clinical factors to improve the prediction models

Whilst most clinical parameters did not display strong separation between the High-WL and Low-WL groups (Table 1), we still hypothesized that the combination of easily accessible clinical parameters could predict weight loss. We constructed a prediction model based on 30 anthropometric and clinical factors, but this model did not predict the weight loss status of the samples significantly better than the models that feature randomly selected genes (median AUC = 0.51, max AUC = 0.75, 95% CI [0.60–0.90], P = 0.12) (Fig. S3).

We then hypothesized that adding these 30 anthropometric and clinical factors, one by one, to the input parameters of Lipid and Virus prediction models may yield to improved predictions. The following factors improved the performance of the Lipid models: systolic blood pressure (median AUC = 0.59, max AUC = 0.74, 95% CI [0.63–0.86], $P = 0.037$) and fibrinogen (median AUC = 0.59, max AUC = 0.77, 95% CI [0.67–0.88], $P = 0.049$)

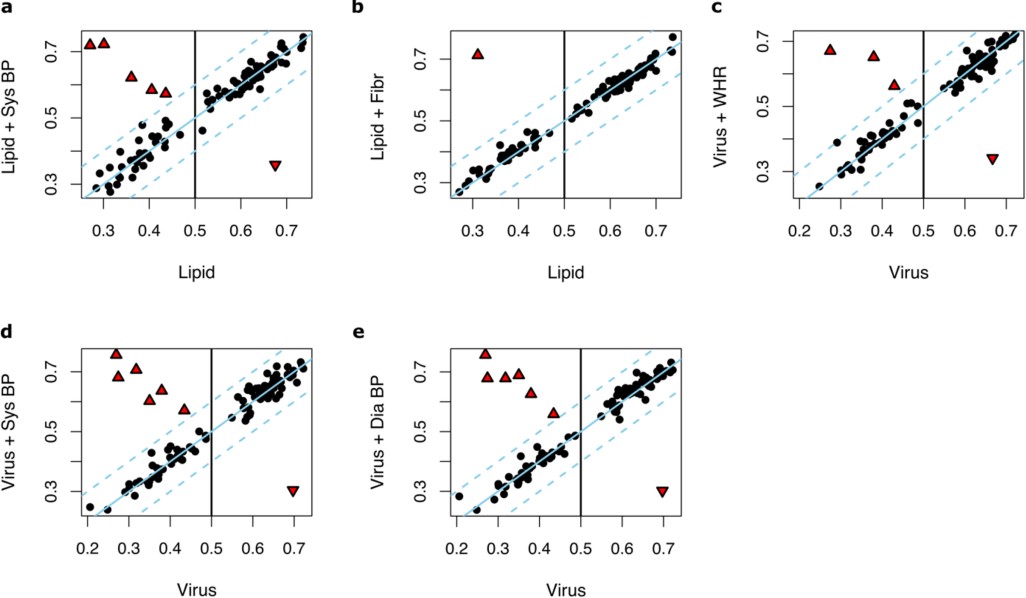

**Figure 4 The effect of incorporating anthropometric and clinical factors into the lipid and virus models.** (A and B) Scatter plots illustrating AUC measurements for weight loss prediction models that as input parameters combine the Lipid genes models and (A) systolic blood pressure and (B) fibrinogen. (C–E) Scatter plots illustrating the AUC measurements for weight loss prediction models that as input parameters combine Virus models and (C) waist-to-hip ratio, (D) systolic blood pressure, and (E) diastolic blood pressure. The Y axis corresponds to the AUC measurements for the combined models whilst the X axis corresponds to the AUCs for the models solely based on lipid metabolism or 'response to virus' gene expression. Dots or triangles indicate individual runs. The black dots represent those runs where the AUC performance differences between the models that the Y axis and X axis represent were less than 0.1. The threshold cutoffs ($|Y-X| = 0.1$) are shown with dashed lines. The solid line represents $Y = X$. The red triangles show the runs in which the difference of the hybrid and original models are significant different ($|Y-X| > 0.1$). Sys BP, Systolic blood pressure; Fibr, fibrinogen; WHR, waist-to-hip ratio; Sys BP, systolic blood pressure.

(Table S1). The performance of the Virus models was improved by adding systolic blood pressure (median AUC = 0.61, max AUC = 0.76, 95% CI [0.65–0.86], $P = 0.018$), diastolic blood pressure (median AUC = 0.59, max AUC = 0.74, 95% CI [0.65–0.86], $P = 0.037$) or waist-to-hip ratio (median AUC = 0.59, max AUC = 0.74, 95% CI [0.67–0.88], $P = 0.037$) (Table S2).

Notably, in most of the runs, no significant changes of performance were observed upon adding anthropometric and clinical factors ($\Delta$AUC < 0.1) (Figs. 4A–4E). However, in one to six out of 100 runs where the Lipid and Virus models had performed poorly (AUC < 0.5), adding these parameters significantly improved the prediction model performances. In contrast, in a few runs, these parameters worsened the prediction performances where the Lipid and Virus models solely based on genes had performed well (AUC > 0.5). Overall, these results indicate that the effect of incorporating anthropometric and clinical factors into the weight-loss prediction model is negligible and should be carried out with caution.

## DISCUSSION

We aimed to study whether baseline gene expression in subcutaneous adipose tissue can predict the weight loss achieved following a caloric restriction and weight maintenance period. By using supervised machine learning, we show that lipid metabolism and 'response to virus' related gene expression in subcutaneous adipose tissue at baseline can predict the weight-loss status (*i.e.*, high-WL or low-WL) of individuals with obesity. Adding clinical variables as input parameters did not necessarily improve the Lipid or Virus prediction models.

In this study, we show that the combined expression of ten lipid metabolism related genes (Lipid models) can predict the weight-loss status following long-term (*i.e.*, 8 months) intervention up to a maximum AUC of 0.74. These models featured the genes including *GPAM*, *DGAT2* and *ELOVL6* which are involved in triacylglycerol synthesis and elongation (*Morgan-Bathke et al., 2015*; *Matsuzaka, 2021*). The high-WL group showed relatively lower baseline expression across all these lipid metabolism genes, which indicate that individuals with a more active triacylglycerol formation in subcutaneous adipose tissue at baseline may have a worse chance of achieving long-term weight loss. This is consistent with our previous work whereby alterations in lipid metabolism genes were observed in differentiating glycemic responders from non-responders (*Valsesia et al., 2020*). Here, we extend these findings by showing that the baseline expression of lipid metabolism genes can also predict the weight loss status of individuals with obesity. However, additional studies are needed to fully elucidate the underlying mechanisms of adipose tissue lipid turnover in weight loss success.

The prediction models composed of the Virus pathways genes were also capable of predicting the weight loss groups with a performance as high as AUC of 0.72. We recognized that the group of genes is broad and heterogenous in the Virus models with half of these genes overlapping with the Lipid metabolism pathway genes (*i.e.*, *ACSL1, CYP3A4, CYP3A5, CYP1A1 and PLA2G4C*). In fact, the performance of Virus models was highly dependent on the overlapping genes. These genes are related to metabolizing xenobiotics and endogenous molecules including steroid hormones and poly-unsaturated fatty acids (*Bishop-Bailey et al., 2014*). Indeed, obesity is known to dysregulate the CYP epoxygenase pathway and evoke a marked suppression of adipose-derived epoxyeicosatrienoic acids levels, thereby affecting adipogenesis (*Zha et al., 2014*). Individuals with less active adipogenesis genes at baseline may have a better chance of achieving long-term weight loss. Therefore, an important role for adipose tissue lipid metabolism in weight loss success is emphasized by these results.

We tried to improve the performance by incorporating multiple anthropometric and clinical factors into the Lipid and Virus prediction models. Although the incorporation of a few parameters (*i.e.*, systolic and diastolic blood pressure, fibrinogen and/or waist-to-hip ratio) significantly improved the performances of Lipid and Virus models in a few runs, overall, they did not affect the performance of the Lipid and Virus models in most runs. Furthermore, the prediction models solely based on the anthropometric and clinical factors did not predict significantly better than models based on randomly selected genes,

which is in line with our previous results (*Valsesia et al., 2020*). Therefore, the performances of the models were not noticeably enhanced, and this once again illustrates the difficulty in predicting weight maintenance based on initial phenotype (*Varkevisser et al., 2019*).

Our results are strengthened by the fact that the DiOGenes study is one of the largest studies of its kind. In order to prevent any overfitting of the prediction models to the studied data, we applied a five-fold cross-validation with 100 resampling. Even though the individuals are deeply phenotyped, the number of included individuals in this study may still be limited for an effective data integration and prediction analysis. Furthermore, we attempted to understand the limits of our prediction models and the dependency of the performance of our discovered prediction models on the training data. The Lipid and Virus models performed the best when their training data included less samples from high-WL individuals who gained weight considerably (>4%) during the weight-maintenance phase, even though during the LCD they had lost weight to an extent that their overall weight loss during the whole 8-months intervention was higher than the median weight loss. This suggests that perhaps some of the individuals who were labelled as high-WL should have been labelled as low-WL or NA. These individuals may have been detected as High-WL due to the limitations of the data rather than their weight loss. As their weight is rising during the weight maintenance phase, we can speculate that they may have been labelled as low-WL if the weight of the studied individuals were continued to be measured for a longer period than 8 months. Furthermore, with our data it is almost impossible to use a more accurate method to label high-WL and low-WL samples (*e.g.*, considering the rate of the weight regain of the studied individuals during the weight maintenance period) as only weight measurements at three time points are available. Another limitation of the current study is that we have classified the participants solely on changes in body weight during the whole dietary intervention and have attempted to predict the weight loss outcome using gene expression in subcutaneous adipose tissue and baseline clinical information. Furthermore, we did not consider the different macronutrient contents of the diets in our analyses since the differences in SAT gene expression was previously reported to be associated with weight variations rather than the dietary macronutrient content (*Márquez-Quiñones et al., 2010*). However, we fully acknowledge that the regulation of body weight is multifactorial and can be influenced by many other factors including lifestyle/behavioral and environmental factors (*Varkevisser et al., 2019*), for which detailed data is lacking in our study.

## CONCLUSIONS

In conclusion, our study demonstrates the importance of phenotyping on molecular level to identify possible mechanisms involved in the dynamics of weight loss success. We show that prediction models based on adipose tissue gene expression related to lipid metabolism, and 'response to virus' (with genes that are also highly associated with lipid metabolism) can predict the weight loss outcome following a dietary intervention. Furthermore, the performance of the prediction models based on 'response to virus' genes were highly dependent on those genes that are also annotated with lipid metabolism. Therefore, our

results confirm the importance of lipid metabolism in subcutaneous adipose tissue in the long-term weight loss of individuals with obesity. Finally, our methodologies and findings can also benefit clinicians and scientists by providing a better understanding of the large inter-individual variability in response to dietary interventions and methods to tackle challenges they introduce.

## ACKNOWLEDGEMENTS

We thank the Institute for Molecular Medicine Finland (FIMM) for providing and maintaining the computing resources on which our analysis was executed.

### Funding

The Diogenes study was supported by the European Commission, the Food Quality and Safety Priority of the Sixth Framework Program (FP6-2005-513946). Birgitta W. van der Kolk was supported by the Finnish Diabetes Research Foundation. Kirsi H. Pietiläinen was funded by the Academy of Finland (grant numbers 335443, 314383, 272376 and 266286), Sigrid Jusélius Foundation, the Academy of Finland Center of Excellence in Research on Mitochondria, Metabolism and Disease (FinMIT; grant number 272376), the Finnish Medical Foundation, the Gyllenberg Foundation, the Novo Nordisk Foundation (grant numbers NNF20OC0060547, NNF17OC0027232 and NNF10OC1013354), Gyllenberg Foundation, the Finnish Diabetes Research Foundation, the Finnish Foundation for Cardiovascular Research, Government Research Funds, the University of Helsinki and Helsinki University Hospital. The funders had no role in study design, data collection and analysis, decision to publish, or preparation of the manuscript.

### Grant Disclosures

The following grant information was disclosed by the authors:
European Commission, the Food Quality and Safety Priority of the Sixth Framework Program: FP6-2005-513946.
Finnish Diabetes Research Foundation.
Academy of Finland: 335443, 314383, 272376 and 266286.
Sigrid Jusélius Foundation.
Academy of Finland Center of Excellence in Research on Mitochondria, Metabolism and Disease (FinMIT): 272376.
Finnish Medical Foundation.
Gyllenberg Foundation.
Novo Nordisk Foundation: NNF20OC0060547, NNF17OC0027232 and NNF10OC1013354.
Gyllenberg Foundation.
Finnish Diabetes Research Foundation.
Finnish Foundation for Cardiovascular Research.

Government Research Funds.
University of Helsinki and Helsinki University Hospital.

## Competing Interests

At the time of the study, Armand Valsesia was a full-time employee at Nestlé Institute of Health Sciences SA. W. H. Saris discloses the receipt of research support from several food companies such as Nestlé, DSM, Unilever, Nutrition et Sante and Danone as well as pharmaceutical companies including GSK, Novartis and Novo Nordisk; he is also an unpaid scientific advisor for the International Life Science Institute (ILSI) Europe. Arne Astrup discloses grants and personal fees received from Gelesis, USA; personal fees from Acino, Switzerland; BioCare Copenhagen, DK; Dutch Beer Institute, NL; Groupe Éthique et Santé, France; IKEA Food Scientific Health Advisory Board, SE; McCain Foods Limited, USA; Navamedic, DK; Novo Nordisk, DK; Pfizer, USA; Saniona, DK; Weight Watchers, USA & Zaluvida, Switzerland; and grants from DC-Ingredients Denmark, which lie beyond the scope of the work reported here. Ellen E. Blaak received financial support from food industry, such as DSM, Danone, Friesland Campina, Avebe and Sensus, partly within the context of public–private consortia and has received funding from pharmaceutical companies including Novartis. She is involved in several task forces and expert groups related to ILSI Europe.g interests.

## Author Contributions

- Ali Oghabian conceived and designed the experiments, performed the experiments, analyzed the data, prepared figures and/or tables, authored or reviewed drafts of the article, and approved the final draft.
- Birgitta W. van der Kolk conceived and designed the experiments, performed the experiments, analyzed the data, prepared figures and/or tables, authored or reviewed drafts of the article, and approved the final draft.
- Pekka Marttinen conceived and designed the experiments, authored or reviewed drafts of the article, and approved the final draft.
- Armand Valsesia conceived and designed the experiments, authored or reviewed drafts of the article, and approved the final draft.
- Dominique Langin conceived and designed the experiments, authored or reviewed drafts of the article, and approved the final draft.
- W. H. Saris conceived and designed the experiments, authored or reviewed drafts of the article, and approved the final draft.
- Arne Astrup conceived and designed the experiments, authored or reviewed drafts of the article, and approved the final draft.
- Ellen E. Blaak conceived and designed the experiments, authored or reviewed drafts of the article, and approved the final draft.
- Kirsi H. Pietiläinen conceived and designed the experiments, authored or reviewed drafts of the article, and approved the final draft.

## Human Ethics

The following information was supplied relating to ethical approvals (*i.e.*, approving body and any reference numbers):

Local ethics committees approved the study, each participant provided written informed consent and the study was carried out in accordance with the principles of the Declaration of Helsinki. Committees included (1) Medical ethical commission from Maastricht University, NL (2) Copenhagen ethical research commission, DK (3) Bedfordshire local Research Ethics Committee, Luton and Dunstable Hospital NHS Trust, UK (4) Ethics Committee of the Faculty Hospital, Prague University, CZ (5) Ethical Commission by NMTI, Sofia, BG (6) Ethical Commission University Potsdam, D (7) Ethical Commission Medical University, Navarra, SP (8) Scientific council Heraklion general university hospital, Heraklion, GR and (9) Commission Cantonale d' éthique de la recherche sur l' être humain, Canton de Vaud, CH.

## Data Availability

The data are available at GEO: GSE95640.

## Clinical Trial Registration

The following information was supplied regarding Clinical Trial registration:

ClinicalTrials.gov number: NCT00390637.

## Supplemental Information

Supplemental information for this article can be found online at http://dx.doi.org/10.7717/peerj.15100#supplemental-information.

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
