# Peer review of "Baseline gene expression in subcutaneous adipose tissue predicts diet-induced weight loss in individuals with obesity"

_PeerJ, doi:10.7717/peerj.15100_

## Round 0.1 · original submission · Major Revisions

As the authors will realize, the reviewers highlight some major issues associated with the experimental design of the study as well as validity of the findings. Therefore, I would like to invite the authors to respond the reviewers' comments.

Reviewer 1 ·

Basic reporting

Yes

Experimental design

Methods-
• Lines 91-92: please specify if this is a secondary analysis of baseline data from subjects who participated in the DiOGenes study
• Lines 167-195: A figure illustrating the methodology would be a helpful addition to better explain the process.

Validity of the findings

Results:
• Line 223-226: p values between the two groups not included in the table
• Fig 1. A hard to read text in the illustrations
• S. Table 1 and S. Table 2 – highlight significant p-values and make it easier to track- needs better organization in a word document
• Lines 267-269: Can authors explain why the prediction models perform poorly when they include a higher amount of data from High -WLwho regained weight significantly?
• Secondly, if the predictive model performs poorly under the above circumstances, how can the model be used a predictive tool be used to determine weight loss maintenance?
Discussion
• Lines 307-308 Did they mean to say short-term intervention instead of long-term?
• Lines 322-324-even though results show that virus pathway genes can predict weight loss performance and overlap with genes from lipid pathway, Fig 1b does not show expression of any other remainder of virus pathway genes in either HIGH or low -WL groups. Could the authors explain the findings? Do virus pathway genes have any significance independent of their overlap with lipid pathway genes?

Additional comments

• The authors have thoroughly acknowledged the strengths and limitations

Reviewer 2 ·

Basic reporting

Minor issues:
1. Table 2 does not seem to be very informative and probably should be moved to supplementary data. Some table with more detailed information on selected genes would be much more helpful.
2. Figure 1a is unreadable
3. The reference information for similarity matrix of the GO categories is not complete – line 181

Experimental design

The authors developed a prediction model based on a selection of genes that were differentially expressed in two groups of individuals: with low weight loss and with high weight loss. In each run the authors extracted significant DEGs and check for enriched biological processes. To build the model the authors selected most frequently appearing genes from three most frequently discovered processes: fatty acid metabolic pathways (discovered in 42 runs), mitosis/meiosis pathways (discovered in 36 runs) and ‘response to virus pathways’ (discovered in 27 runs).
1. I am not fully convinced by the genes’ selection method:
a. The authors should provide information on how often each selected gene was present in DEG set. Considering that sets of genes in enriched categories differed between runs, some of selected genes could demonstrate significant changes in the expression in relatively small number of runs. For example, the ‘response to virus’ category was discovered in 27 runs. Therefore, even if a gene was present in DEG set in half of these runs it would be just 13-14 instances out of 100.
b. Selecting genes only from enriched in DEG categories seems to be limiting, especially when no category was discovered in at least half of runs. Did the authors check if there are any other genes that are frequently in DEG set although do not belong to any enriched GO category? Could the authors provide some data on this? Did the authors check if any other frequently discovered genes could improve the model?
c. Median AUC for each GO category is not very high (below 0.6). Did the authors check if some mixed model, i.e. combinations of selected genes from all three categories, would perform better?
2. The authors demonstrated that runs in which the training set included more High-WL individuals who gained the weight performed worse. Only 7 individuals gained more than 4% so this is not a high number. However, even some subset of these individuals negatively influenced the outcome of the run. Shouldn’t these individuals be excluded?

Validity of the findings

The information on RNA-seq data accession numbers from appropriate repository is not provided.

Additional comments

no comment

---

## Round 0.2 · accepted · Accept

Based on my own evaluation as Academic Editor, the authors have successfully addressed all comments raised by the reviewers. The manuscript can now be accepted as it is. I congratulate the authors for their work.